# Distilling Datasets Into Less Than One Image

**Asaf Shul**[*]                                                                           *asaf.shul@mail.huji.ac.il*
*The Hebrew University of Jerusalem*

**Eliahu Horwitz**[*]                                                                 *eliahu.horwitz@mail.huji.ac.il*
*The Hebrew University of Jerusalem*

**Yedid Hoshen**                                                                    *yedid.hoshen@mail.huji.ac.il*
*The Hebrew University of Jerusalem*

**Reviewed on OpenReview:** *https://openreview.net/forum?id=qsipSdfWeV&invitationId*[†]

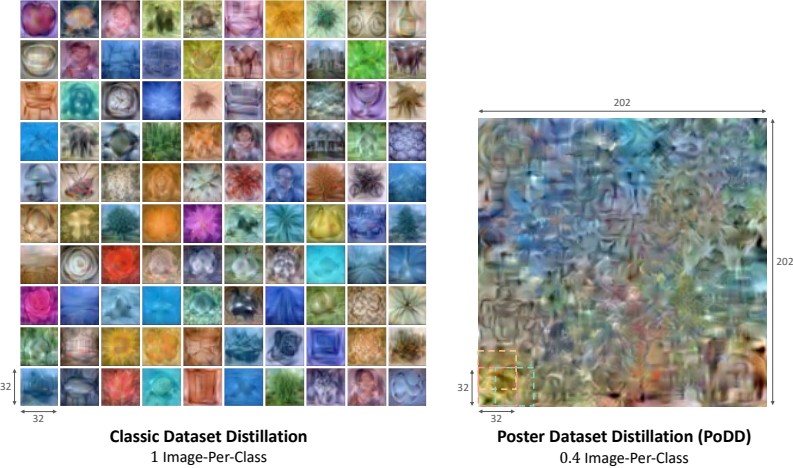

Figure 1: ***Poster Dataset Distillation (PoDD):*** We propose PoDD, a new dataset distillation setting for a tiny, under 1 image-per-class (IPC) budget. In this example, the standard method attains an accuracy of 35.5% on CIFAR-100 with approximately $100k$ pixels, PoDD achieves an accuracy of 35.7% with less than half the pixels (roughly $40k$)

## Abstract

Dataset distillation aims to compress a dataset into a much smaller one so that a model trained on the distilled dataset achieves high accuracy. Current methods frame this as maximizing the distilled classification accuracy for a budget of $K$ distilled *images-per-class*, where $K$ is a positive integer. In this paper, we push the boundaries of dataset distillation, compressing the dataset into less than an image-per-class. It is important to realize that the meaningful quantity is not the number of distilled images-per-class but the number of distilled pixels-per-dataset. We therefore, propose *Poster Dataset Distillation* (PoDD), a new approach that distills the entire original dataset into a single poster. The poster approach motivates new technical solutions for creating training images and learnable labels. Our method can achieve comparable or better performance with less than an image-per-class compared to existing methods that use one image-per-class. Specifically, our method establishes a new state-of-the-art performance on CIFAR-10, CIFAR-100, and CUB200 on the well established 1 IPC benchmark, while using as little as 0.3 images-per-class.

---

[*]Equal contribution
[†]Project page: https://horwitz.ai/podd/

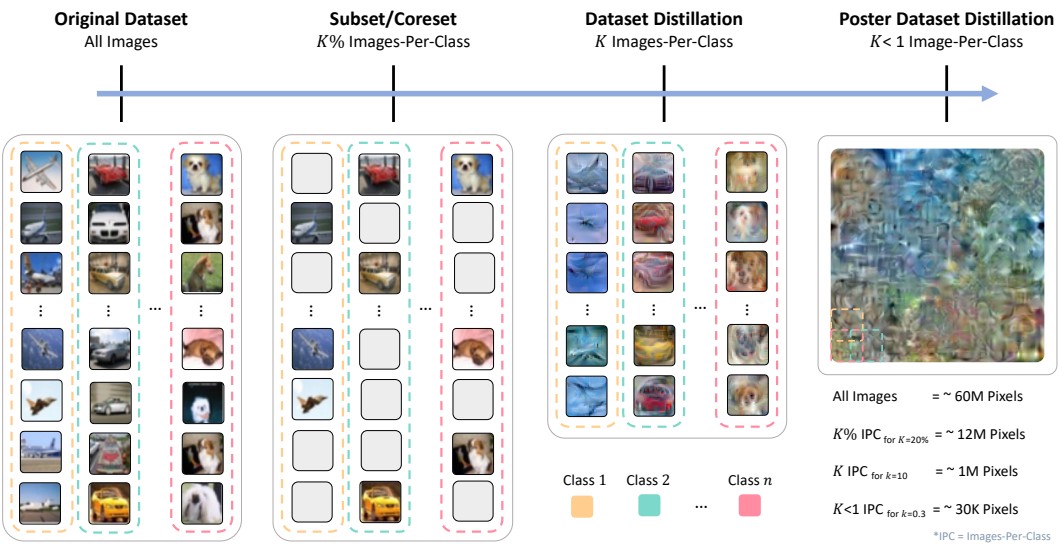

Figure 2: ***Dataset compression scale:*** We show increasingly more compressed methods from left to right. The original dataset contains all of the training data and does not perform any compression. Coreset methods select a subset of the original dataset, without modifying the images. Dataset distillation methods compress an entire dataset by synthesizing $K \in \mathbb{N}^+$ images-per-class (IPC). Our method, ***Poster Dataset Distillation*** (PoDD) distills an entire dataset into a single poster that achieves the same performance as 1 IPC while using as little as 0.3 IPC

## 1 Introduction

Deep-learning methods require large training datasets to achieve high accuracy. Dataset distillation (Wang et al., 2018) allows distilling large datasets into smaller ones so that training on the distilled dataset results in high accuracy. Concretely, dataset distillation methods synthesize the $K$ images-per-class (IPC) that are most relevant for the classification task. Dataset distillation has been very successful, achieving high accuracy with as little as a single image-per-class.

In this paper, we ask: "can we go lower than one image-per-class?" Existing dataset distillation methods are unable to do this as they synthesize one or more distinct images for each class. Assuming there are $n$ classes, such methods would require distilling at least $n$ images. On the other hand, using less than 1 IPC implies that several classes share the same image, which current methods do not allow. We therefore propose ***Poster Dataset Distillation*** (PoDD), which distills an entire dataset into a single larger image, that we call a poster. The benefit of the poster representation is the ability to use patches that overlap between the classes, thus better utilizing redundancies of pixels across images. We can set the size of the poster so it has significantly fewer pixels than in $n$ images, therefore enabling distillation with less than 1 IPC. We find that a correctly distilled poster is sufficient for training a model with high accuracy, outperforming current methods. See Fig. 2 for an overview of different dataset compression methods.

To illustrate the idea of a poster, consider CIFAR-100(Krizhevsky et al., 2009) where each image is of size $32 \times 32$ pixels. Current methods synthesize images independently and thus must use at least 1 IPC (see Fig. 10 (left)). Choosing 1 IPC for CIFAR-100 entails using 100 images, each of size $32 \times 32$ pixels. In contrast, PoDD synthesizes a single poster shared between all classes. During distillation, we optimize all the pixels so that a classifier trained on the resulting dataset achieves maximal accuracy. For example, to achieve 0.4 IPC, we represent the entire dataset as a single poster of size $202 \times 202$ pixels (see Fig. 10 (right)). This has about the same number of pixels as 40 images, each of size $32 \times 32$. The number of effective IPCs is therefore directly given by the size of the poster.

To distill a poster, we first initialize all pixels with random values. We transform the poster into a dataset in a differentiable way, by extracting overlapping patches, each with the same size as an image in the source

dataset (e.g., $32 \times 32$ for CIFAR-100). During distillation, we optimize this set of overlapping poster patches and propagate the (overlapping) accumulated gradients from the distillation algorithm back to the poster. The optimization objective is to synthesize a poster such that a classifier trained on the dataset extracted from it will reach high classification accuracy. This process requires a label for every patch, we therefore propose ***P**oster **D**ataset **D**istillation **L**abeling* (PoDDL), a method for poster labeling that supports both fixed and learned labels.

Since classes can now share pixels, their order within the poster matters as it implies which classes share pixels with each other. It is thus important to find a beneficial ordering of classes on the poster. To this end, we propose ***P**oster **C**lass **O**rdering* (PoCO), an algorithm that uses CLIP (Radford et al., 2021) text embeddings to order the classes semantically and efficiently.

Overall, in addition to stretching the limits of dataset compression, the overlapping patches in PoDD formulation lead to improved performance on existing 1 IPC benchmarks. This holds true even when using less than 1 IPC, where PoDD can match or improve the performance that previous methods achieved on the well established 1 IPC benchmark. Indeed, sometimes PoDD can outperform competing methods using as low as 0.3 IPC. PoDD sets a new state-of-the-art for 1 IPC on CIFAR-10, CIFAR-100, and CUB200.

To summarize, our main contributions are:

1. Proposing the poster dataset structure, which extends dataset distillation for tiny, less than 1 IPC budgets.

2. Developing a method to perform PoDD that constitutes of a class ordering algorithm (PoCO) and a labeling strategy (PoDDL).

3. Performing extensive experiments that demonstrate the effectiveness of PoDD with as low as 0.3 IPC and achieving a new SoTA on the well established 1 IPC benchmark.

## 2 Related works

Dataset distillation, introduced by Wang et al. (2018), aims to compress an entire dataset into a smaller, synthetic one. The goal is that methods trained on the distilled dataset will achieve similar accuracy to a model trained on the original dataset. As highlighted by (Rebuffi et al., 2017; Castro et al., 2018; Jubran et al., 2019), dataset distillation shares similarities with coreset selection. Coreset selection identifies a representative subset of samples from the training set that can be used to train a model to the same accuracy. Unlike coreset selection, the generated synthetic samples of dataset distillation provide flexibility and improved performance through continuous gradient-based optimization techniques. Dataset distillation methods can be categorized into 4 main groups: i) *Meta-Model Matching* (Wang et al., 2018; Nguyen et al., 2021; Loo et al., 2022; Zhou et al., 2022; Feng et al., 2023) minimize the discrepancy between the transferability of models trained on a distilled data and those trained on the original dataset. ii) *Gradient Matching* (Zhao et al., 2020; Zhao & Bilen, 2021; Lee et al., 2022b; Kim et al., 2022), proposed by Zhao et al. (2020), performs one-step distance matching between a network trained on the target dataset and the same network trained on the distilled dataset. This avoids unrolling the inner loop of Meta-Model Matching methods. iii) *Trajectory Matching*(Cazenavette et al., 2022; Cui et al., 2023), proposed by Cazenavette et al. (2022), focuses on matching the training trajectories of models trained on the target distilled dataset and the original dataset. iv) *Distribution Matching*(Zhao & Bilen, 2023; Wang et al., 2022; Lee et al., 2022a), introduced by Zhao & Bilen (2023), solves a proxy task via a single-level optimization, directly matching the distribution of the original dataset and the distilled dataset. See (Sachdeva & McAuley, 2023) for an in-depth explanation and comparisons of the various distillation methods. Common to all these methods is the use of at least one IPC. Deng & Russakovsky (2022) observed that shared representations facilitate better distillation. In contrast to their approach, in this paper we introduce a new shared structure which we call a poster, allowing distilling a dataset into less than 1 IPC.

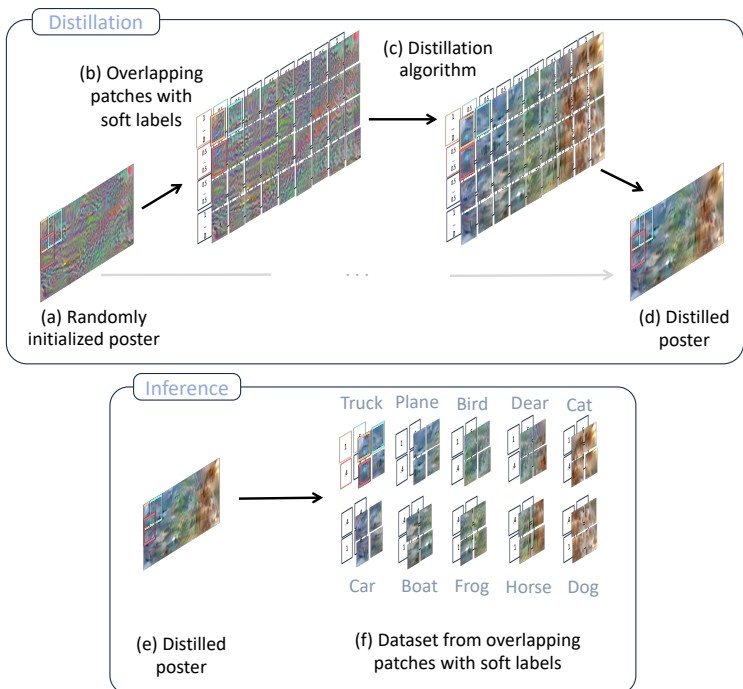

Figure 3: **_PoDD overview:_** We propose PoDD, a new dataset distillation setting for under 1 images-per-class. We start by initializing a random poster (a), during distillation, we optimize overlapping patches and soft labels (b-c). The final distilled poster has fewer pixels than the combined pixels of the individual images (d). During inference, we extract overlapping patches and soft labels from the distilled poster and use them to train a downstream model (e-f). PoDD achieves comparable or better accuracy to current methods while using as little as a third of the pixels

## 3   Preliminaries

Many methods tackle dataset distillation as a bi-level optimization problem. In this setup, the inner loop essentially involves training a model with weights $\theta$ on a *distilled* dataset $\mathcal{D}_{\text{syn}}$. The outer loop optimizes the pixels of the distilled dataset $\mathcal{D}_{\text{syn}}$, so a model trained on $\mathcal{D}_{\text{syn}}$ has the maximal accuracy on the *original* dataset $\mathcal{D}$. Let $\mathcal{L}_{\mathcal{D}}(\theta)$ denote the average value of the objective function, of model $\theta$ on the dataset $\mathcal{D}$. Formally the optimization problem can be described as follows:

$$\underbrace{\arg\min_{\mathcal{D}_{\text{syn}}} \ \mathcal{L}_{\mathcal{D}}\left(\theta^*\right)}_{\text{Outer loop}} \quad \text{s.t} \quad \theta^* = \underbrace{\arg\min_{\theta} \ \mathcal{L}_{\mathcal{D}_{\text{syn}}}(\theta)}_{\text{Inner loop}} \tag{1}$$

We consider the case where the inner-loop optimization consists of $T_{end}$ SGD steps. The most common solution is backpropagation through time (BPTT) which unrolls the inner loop SGD optimization for $T_{end}$ steps. It then uses computationally demanding backpropagation calculation to compute the gradient of the loss with respect to the distilled dataset $\mathcal{D}_{\text{syn}}$,

$$\underbrace{\arg\min_{\mathcal{D}_{\text{syn}}} \ \mathbb{E}_{\theta_0 \sim P_{\theta}} \left[\mathcal{L}_{\mathcal{D}}\left(\theta_{T_{end}}\right)\right]}_{\text{Outer loop}} \quad \text{s.t} \quad \underbrace{\theta_{t+1} \leftarrow \theta_t - \eta \cdot \nabla_{\theta} \mathcal{L}_{\mathcal{D}_{\text{syn}}}(\theta_t)}_{\text{Inner loop unrolling}} \tag{2}$$

Backpropagating through $T_{end}$ timesteps is infeasible for large values of $T_{end}$, therefore many methods propose ways to reduce the computational costs. Here, we use RaT-BPTT (Feng et al., 2023), which computes the inner loop through a random number of SGD steps $T_{end} \sim Random[\Delta T, \Delta T + 1, ..., T]$. It then approximates the gradient with respect to the distilled dataset by only backpropagating through the

final $\Delta T << T$ steps. We chose RaT-BPTT because it achieves the top performance across many dataset distillation benchmarks.

## 4 PoDD: Poster Dataset Distillation

Our goal is to perform dataset distillation with less than 1 IPC. Our main insight is that sharing pixels between classes can be effective, as this would make better use of redundant pixels. Clearly, this requires more than one class to share an image. In this section, we propose Poster Dataset Distillation (PoDD) which provides the methods to realize this idea.

### 4.1 A shared poster representation

The key idea in this work is to distill an entire dataset into a single larger image that we call a poster. The poster can be of arbitrary height $d_h$ and width $d_w$, leading to a total of $d_h \times d_w$ pixels; posters with fewer pixels are said to be more *compact*. Furthermore, we define a fixed procedure for converting a poster into a distilled dataset. Our procedure extracts multiple overlapping patches at a fixed, user determined stride; each extracted patch is equal in size to the original images (see Fig. 3 for an overview). We employ the same stride for both rows and columns and denote the total number of extracted patches by $p$. Consequently, this yields a dataset of images, some of which share pixels. Without any overlap, optimizing the poster is equivalent to distilling each image separately.

We initialize the poster with standard Gaussian noise and optimize its pixels end-to-end through both the dataset expansion described above and the inner-loop bi-level optimization described in 3. As a result, the gradients from overlapping patches are accumulated. The size of the distilled dataset is measured by the total number of pixels in the poster, allowing us to compare with previous approaches that used one or more IPC. We provide pseudocode for PoDD in 2.

### 4.2 PoCO: Poster Class Ordering

The poster representation relies on shared pixels between neighboring classes. To maximize the effectiveness of the shared pixels, it is therefore important to establish a good structure of neighboring classes. We hypothesize that a poster would be more effective when pixels are shared between semantically related classes (e.g., Man, Woman, Boy vs. Plate, Tree, Bus).

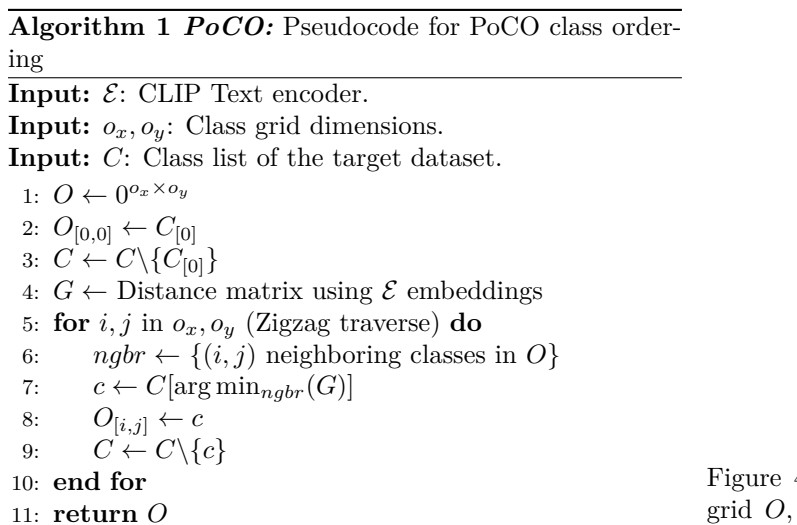

**Algorithm 1 *PoCO:* Pseudocode for PoCO class ordering**

**Input:** $\mathcal{E}$: CLIP Text encoder.
**Input:** $o_x, o_y$: Class grid dimensions.
**Input:** $C$: Class list of the target dataset.
1: $O \leftarrow 0^{o_x \times o_y}$
2: $O_{[0,0]} \leftarrow C_{[0]}$
3: $C \leftarrow C \backslash \{C_{[0]}\}$
4: $G \leftarrow$ Distance matrix using $\mathcal{E}$ embeddings
5: **for** $i, j$ in $o_x, o_y$ (Zigzag traverse) **do**
6: $\quad ngbr \leftarrow \{(i,j)$ neighboring classes in $O\}$
7: $\quad c \leftarrow C[\arg\min_{ngbr}(G)]$
8: $\quad O_{[i,j]} \leftarrow c$
9: $\quad C \leftarrow C \backslash \{c\}$
10: **end for**
11: **return** $O$

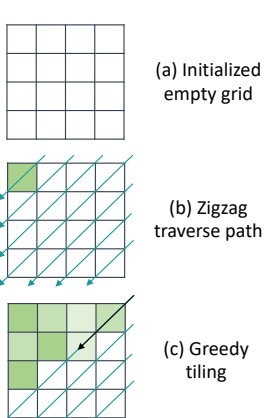

(a) Initialized empty grid

(b) Zigzag traverse path

(c) Greedy tiling

Figure 4: *PoCO tiling:* (a) Initialize empty grid $O$, (b) Traverse $O$ in a Zigzag order, (c) Greedily tile $O$ using CLIP text embeddings

We propose a method for approximating a good initial class neighborhood structure. First, we extract an embedding for each class name using the CLIP (Radford et al., 2021) text-encoder $\mathcal{E}$. Given the set of embeddings, we calculate the pairwise distance matrix between all classes. Using this pairwise class distance, we design a greedy procedure for placing the classes on a rectangular grid $O^{o_x \times o_y}$, where $o_x \cdot o_y = |C|$, denoting the spatial positions of classes. Let $C$ be the set of classes and $C_p$ be the set of already placed classes. We traverse the grid in a Zigzag order (Chai et al., 2017), and at each step place a class as follows:

$$O_{i,j} = \min_{c \in C \setminus C_p} \left( \sum_{m \in \{O_{i-1,j}, O_{i,j-1}\}} \left( 1 - \frac{\mathcal{E}(m) \cdot \mathcal{E}(c)}{||\mathcal{E}(m)|| \cdot ||\mathcal{E}(c)||} \right) \right), \quad i \in [o_x], \;\; j \in [o_y]$$

Intuitively, at each step, we place a remaining class that has the lowest distance from all its already-placed neighbors. We visualize this Zigzag traversal in Fig. 4 and summarize the PoCO algorithm in 1, in Fig. 7 we show an example PoCO tiling for CIFAR-100.

### 4.3 PoDDL: Poster Dataset Distillation Labeling

Having initialized the poster and the class order matrix $O$, we now describe our labeling strategy. Previous approaches use one or more images-per-class, hence, they can simply assign a single label per image. However, using a single label for the entire poster is not a good option, instead, we assign a soft label (Sucholutsky & Schonlau, 2021) vector to each overlapping patch. We therefore design a poster-oriented soft labeling strategy that supports both fixed and learned labels, see Fig. 5 for an overview.

**Fixed labels.** We upsample the class order matrix $O$ to the size of the poster $d_h \times d_w$. For each overlapping patch, we extract its corresponding class label window. We compute its majority class and use it as the one-hot label for the patch. In the case of ties, we use a soft label with equal probabilities for the majority classes.

**Learned labels.** As our method extracts an arbitrary number of overlapping patches, learning a soft label for each patch would require more parameters than previous approaches. To keep the number of parameters constant, we learn a parameter tensor of the same size as previous works, and interpolate it to each overlapping window.

Concretely, we learn a label tensor $Y$ of size $o_x \times o_y \times n$ and spatially upscale it to the shape of the poster using nearest neighbor interpolation. The final size of $Y$ is $d_h \times d_w \times n$. For each overlapping patch, we extract the corresponding label window and average pool it. To achieve a valid label distribution, we $L_1$ normalize the resulting vector. We use this vector as the learned soft label of the window. After each gradient step, we clip negative values of $Y$ to zero to avoid negative probabilities. Unlike the fixed labels, the learned labels are optimized alongside the distillation process.

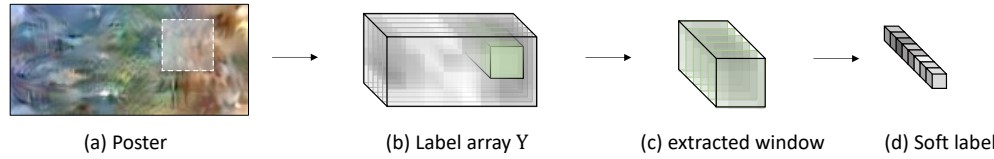

(a) Poster      (b) Label array Y      (c) extracted window      (d) Soft label

Figure 5: **_PoDDL extraction:_** Each poster patch has a corresponding patch in the label array (a-b). We compute the poster patch label by extracting a patch along the channels of the label array (c). To obtain the final soft label for a given poster patch, we pool and normalize the extracted label window, resulting in a soft label vector (d). PoDDL supports both fixed and learned labels

---

**Algorithm 2** *PoDD:* Pseudocode using PoDDL learned labels

---

**Input:** *Alg*: Distillation algorithm.
**Input:** $\mathcal{D}$: Dataset with classes $C$.
**Input:** $d_h$, $d_w$: Poster size.
**Input:** $p$: Number of overlapping patches.
  1: $\mathcal{P} \sim \mathcal{N}(0,1)^{d_x \times d_w}$                                 # Initialize a poster from Gaussian

  2: $O \leftarrow \text{PoCO}(C)$                                         # Initialize class order
  3: $Y \leftarrow \text{PoDDL}(O, p)$                                  # Initialize distilled labels array
  4: **for** each distillation step **do**
  5:     $\mathcal{D}_{\text{syn}} := \{(p, l) \text{ overlapping patches and labels from } \mathcal{P} \text{ and } Y\}$
  6:     $\mathcal{P}, Y \leftarrow Alg(\mathcal{D}, \mathcal{D}_{\text{syn}})$                        # Distill one step
  7: **end for**
  8: Return $\mathcal{P}$, $Y$

---

# 5 Experiments

## 5.1 Experimental setting

**Datasets.** We evaluate PoDD on four datasets commonly used to benchmark dataset distillation methods: i) *CIFAR-10:* 10 classes, $50k$ images of size $32 \times 32 \times 3$ (Krizhevsky et al., 2009). ii) *CIFAR-100:* 100 classes, $50k$ images of size $32 \times 32 \times 3$ (Krizhevsky et al., 2009). iii) *CUB200:* 200 classes, $6k$ images of size $32 \times 32 \times 3$ (Welinder et al., 2010). iv) *Tiny-ImageNet:* 200 classes, $100k$ images of size $64 \times 64 \times 3$ (Le & Yang, 2015).

**Distillation Method.** For the dataset distillation algorithm, we use RaT-BPTT (Feng et al., 2023), a recent method that achieved SoTA performance on CIFAR-10, CIFAR-100, and CUB200 by a wide margin. In particular, RaT-BPTT's architecture uses three convolutional layers for $32 \times 32$ datasets and four layers for $64 \times 64$ datasets. **Baselines.** Our baselines can be divided into two groups: i) *Inner-Loop:* BPTT (Deng & Russakovsky, 2022), empirical feature kernel (FRePO) (Zhou et al., 2022), and reparameterized convex implicit gradient (RCIG) (Loo et al., 2023), and ii) *Modified Objectives:* gradient matching with augmentations (DSA) (Zhao & Bilen, 2021), distribution matching (DM) (Zhao & Bilen, 2023), trajectory matching (MTT) (Cazenavette et al., 2022), flat trajectory distillation (FDT) (Du et al., 2023), and TESLA (Cui et al., 2023). We use the results reported by the baselines.

**Evaluation.** Following the protocol of (Zhao & Bilen, 2021; Deng & Russakovsky, 2022), we evaluate the distilled poster using a set of 8 different randomly initialized models with the same ConvNet (Gidaris & Komodakis, 2018) architecture used by DSA, DM, MTT, RaT-BPTT, and others. The architecture includes convolutional layers of 128 filters with kernel size $3 \times 3$ followed by instance normalization(Ulyanov et al., 2016), ReLU (Nair & Hinton, 2010) activation, and an average pooling layer with kernel size $2 \times 2$ and stride 2. We report the mean and standard deviation across the 8 random models. We evaluate two IPC regimes:

*Less than one IPC.* We compute the results for IPCs within the range: $K \in [0.3, 0.4, ..., 1)$. Our criterion for success is if PoDD with $K$ IPC performs comparably or better than RaT-BPTT with 1 IPC. To comply with previous baselines, we used PoDDL, which optimized the same number of label parameters as previous methods with 1 IPC.

*One IPC.* To properly compare PoDD with existing methods we also evaluate it with the same total number of pixels used by our baselines (1 IPC). Since PoDD still uses overlapping patches, this evaluates the impact of compressing inter-class redundancies in the shared pixels, this is compared to the baselines which are unable to do so.

**Implementation Details** We use the same distillation hyper-parameters used by RaT-BPTT (Feng et al., 2023) except for the batch sizes. To fit the distillation into a single GPU (we use an NVIDIA A40), we use the maximal batch size we can fit into memory for a given dataset (see exact breakdown below). Since the

Table 1: ***Less than one Image-Per-Class:*** PoDD with less than one IPC (image-per-class) often out-performs state-of-the-art methods with 1-IPC; in some cases with as little as 0.3 IPC. In (red), the relative performance drop compared to the 1 IPC results. In **bold**, the lowest IPC for which PoDD beats the current SoTA

| Method | IPC ↓ | CIFAR-10 ↑ | CIFAR-100 ↑ | CUB200 ↑ | T-ImageNet ↑ |
|---|---|---|---|---|---|
| RaT-BPTT Feng et al. (2023) | 1.0 | $53.2_{\pm0.7}$ | $35.3_{\pm0.4}$ | $13.8_{\pm0.3}$ | $20.1_{\pm0.3}$ |
| PoDD (Ours) | 1.0 | $59.1_{\pm0.5}$ | $38.3_{\pm0.2}$ | $16.2_{\pm0.3}$ | $20.0_{\pm0.3}$ |
| PoDD (Ours) | 0.9 | $58.4_{\pm0.5(1\%)}$ | $37.4_{\pm0.2(2\%)}$ | $15.2_{\pm0.4(6\%)}$ | $19.5_{\pm0.2(2\%)}$ |
| | 0.8 | $56.7_{\pm0.7(4\%)}$ | $37.3_{\pm0.1(3\%)}$ | $15.6_{\pm0.3(4\%)}$ | $19.0_{\pm0.2(5\%)}$ |
| | 0.7 | $\mathbf{54.6}_{\pm0.5(8\%)}$ | $37.0_{\pm0.2(3\%)}$ | $15.0_{\pm0.3(8\%)}$ | $18.6_{\pm0.1(7\%)}$ |
| | 0.6 | $50.6_{\pm0.3(15\%)}$ | $36.6_{\pm0.3(5\%)}$ | $15.1_{\pm0.2(7\%)}$ | $18.8_{\pm0.2(6\%)}$ |
| | 0.5 | $49.5_{\pm0.5(16\%)}$ | $36.0_{\pm0.3(6\%)}$ | $15.0_{\pm0.3(8\%)}$ | $18.7_{\pm0.1(7\%)}$ |
| | 0.4 | $47.1_{\pm0.4(20\%)}$ | $\mathbf{35.7}_{\pm0.2(7\%)}$ | $15.0_{\pm0.2(7\%)}$ | $18.4_{\pm0.2(8\%)}$ |
| | 0.3 | $42.3_{\pm0.3(28\%)}$ | $34.7_{\pm0.2(10\%)}$ | $\mathbf{14.8}_{\pm0.5(9\%)}$ | $18.4_{\pm0.1(8\%)}$ |
| Full Dataset (No Distillation) | All Images | $83.5_{\pm0.2}$ | $55.3_{\pm0.3}$ | $20.1_{\pm0.3}$ | $37.6_{\pm0.5}$ |

optimization is bi-level, distillation methods have two batch types, one for the distilled data which we denote by $bs_d$, and one for the original dataset which we denote by $bs$.

In addition to the $K$ IPC parameter, we need to control the degree of patch overlap. In other words, given a dataset with $n$ classes and after fixing $K$ (i.e., once we fix the size of the poster), we need to decide on the number of overlapping patches to divide the poster into. As the number of classes and image resolution varies between the datasets, we use a different $p$ for each one. Concretely, we use: i) *CIFAR-10*: $p = 96_{(16\times6)}$ patches, $bs_d = 96$, $bs = 5000$, $4k$ epochs. ii) *CIFAR-100*: $p = 400_{(20\times20)}$ patches, $bs_d = 50$, $bs = 2000$, $2k$ epochs. iii) *CUB200*: $p = 1800_{(60\times30)}$ patches, $bs_d = 200$, $bs = 3000$, $8k$ epochs. iv) *Tiny-ImageNet*: $p = 800_{(40\times20)}$ patches, $bs_d = 30$, $bs = 500$, 500 epochs.

We use the learned labels variant of PoDDL for all of our experiments except for CIFAR-10 with $K \in [0.7, 0, 8, 0.9, 1.0]$ IPC in which we use the fixed labels variant (for which learned labels did not provide additional benefit). We use a learning rate of 0.001 for CIFAR-10, CIFAR-100, and CUB200. For Tiny-ImageNet into a single GPU we use a much smaller batch size and a learning rate of 0.0005.

## 5.2 Results

**Less than one IPC.** We now test our initial question: "can we go lower than one image-per-class?" Using PoDD, we show that across all four datasets, we can go much lower than 1 IPC and for 3 out of the 4 datasets even maintain on-par performance to the SoTA baseline. As hypothesized, using a poster that shares pixels between multiple classes allows us to reduce redundancies between classes in the distilled patches (See 1). This effect is intensified when distilling datasets with a large number of classes, e.g., for CIFAR-100 we can use 0.4 IPC and for CUB200 we can use as little as 0.3 IPC and still outperform the baseline method.

**One IPC.** Having shown the feasibility of distilling a dataset into less than one IPC, we now quantitatively evaluate the benefit of the poster representation. To this end, we use the 1 IPC setting which allows us to decouple the pixel count from the pixel sharing. Essentially, we are investigating whether the pixel-sharing in our poster can boost performance, even when the number of pixels matches our baseline. Our method outperforms the state-of-the-art for CIFAR-10, CIFAR-100, and CUB200, setting a new SoTA for 1 IPC dataset distillation (See Tab. 2).

## 5.3 Ablations

**Class order ablation.** We ablate the impact of the class ordering on the performance of PoDD on CIFAR-10. We first compute the distillation performance after 250 distillation steps with 0.3 IPC for 5 random class orderings. The score of each ordering is the inverse of the sum of distances between all neighboring class pairs. The distance matrix is defined in the same way as in PoCO i.e., using the embeddings of the CLIP

Table 2: **One Image-Per-Class:** Performance of PoDD under the 1 image-per-class (IPC) setting compared to SoTA dataset distillation methods across 4 datasets. PoDD sets a new SoTA for CIFAR-10, CIFAR-100, and CUB200. On Tiny-ImageNet, PoDD achieves comparable results to the underlying distillation method it uses (RaT-BPTT). All models used ConvNet architecture with 3 convolution layers for 32 × 32 images and 4 convolution layers for 64 × 64 images (one exception is DSA which uses AlexNet for CIFAR-10 and CIFAR-100 as seen in its original paper)

| Method | CIFAR-10 ↑ | CIFAR-100 ↑ | CUB200 ↑ | T-ImageNet ↑ | Average ↑ |
|---|---|---|---|---|---|
| BPTT_{Deng & Russakovsky (2022)} | $49.1_{\pm0.6}$ | $21.3_{\pm0.6}$ | - | - | - |
| FRePO_{Zhou et al. (2022)} | $45.6_{\pm0.1}$ | $26.3_{\pm0.1}$ | - | $16.9_{\pm0.1}$ | - |
| RCIG_{Loo et al. (2023)} | $49.6_{\pm1.2}$ | $35.5_{\pm0.7}$ | - | $\mathbf{22.4}_{\pm0.3}$ | - |
| RaT-BPTT_{Feng et al. (2023)} | $53.2_{\pm0.7}$ | $35.3_{\pm0.4}$ | $13.8_{\pm0.3}$ | $20.1_{\pm0.3}$ | $30.6_{\pm0.4}$ |
| DSA_{Zhao & Bilen (2021)} | $28.8_{\pm0.7}$ | $13.9_{\pm0.3}$ | $1.3_{\pm0.1}$ | $6.6_{\pm0.2}$ | $12.7_{\pm0.3}$ |
| DM_{Zhao & Bilen (2023)} | $26.0_{\pm0.8}$ | $11.4_{\pm0.3}$ | $1.6_{\pm0.1}$ | $3.9_{\pm0.2}$ | $10.8_{\pm0.3}$ |
| MTT_{Cazenavette et al. (2022)} | $46.3_{\pm0.8}$ | $24.3_{\pm0.3}$ | $2.2_{\pm0.1}$ | $8.8_{\pm0.3}$ | $20.4_{\pm0.4}$ |
| FTD_{Du et al. (2023)} | $46.8_{\pm0.3}$ | $25.2_{\pm0.2}$ | - | $10.4_{\pm0.3}$ | - |
| TESLA_{Cui et al. (2023)} | $48.5_{\pm0.8}$ | $24.8_{\pm0.4}$ | - | - | - |
| PoDD (Ours) | $\mathbf{59.1}_{\pm0.5}$ | $\mathbf{38.3}_{\pm0.2}$ | $\mathbf{16.2}_{\pm0.3}$ | $20.0_{\pm0.3}$ | $\mathbf{33.4}_{\pm0.3}$ |
| Full Dataset (No Distillation) | $83.5_{\pm0.2}$ | $55.3_{\pm0.3}$ | $20.1_{\pm0.3}$ | $37.6_{\pm0.5}$ | $49.1_{\pm0.3}$ |

*Inner Loop* — rows BPTT through RaT-BPTT. *Modified Objectives* — rows DSA through TESLA.

| Number of Patches ($p_x \times p_y$) | Test Accuracy $500_{epochs}$ |
|---|---|
| $10_{(5\times2)}$ | 45.15% |
| $24_{(8\times3)}$ | 47.28% |
| $40_{(10\times4)}$ | 56.77% |
| $60_{(12\times5)}$ | 54.14% |
| $96_{(16\times6)}$ | 56.73% |
| $126_{(18\times7)}$ | 55.55% |
| $160_{(20\times8)}$ | 57.61% |

Figure 6: **Number of patches ablation:** Effect of the patch overlap on the accuracy (CIFAR-10, 1 IPC, 500 distillation steps). Using 10 patches with no patch overlap results in the lowest accuracy. When increasing the number of patches beyond 24, the results improve significantly. added titles to the axes

text encoder. We find that the class ordering can indeed impact the performance of the distilled poster, with a correlation coefficient of 0.76 between the score of the ordering and the accuracy the distilled poster achieves. This correlation motivates PoCO's search for a good initial class ordering rather than using a random one.

**Patch number ablation.** We ablate the role of the amount of overlap between patches of the poster (i.e., the number of patches for a given poster size and dataset). To study this, we use CIFAR-10 at 1 IPC, we run PoDD for 500 steps multiple times, each with a progressively increasing number of patches. As can be seen in Fig. 6, using the same number of patches as the number of classes (i.e., no overlap between patches) results in the lowest score; this is expected as this is exactly the RaT-BPTT baseline. When increasing the number of patches, we observe that beyond a certain patch number threshold, the results improve drastically. This demonstrates the significance of the poster representation and the use of overlapping patches. Since the number of patches has a direct effect on the distillation time and the training time of downstream models, we use a small number of patches for the larger datasets and a larger number of patches for the smaller datasets.

**More than 1-IPC ablation.** Our evaluation focused on 1 or fewer IPC, a setting that has a performance gap compared to models trained on the full dataset. We prioritized this low IPC setting as it represents

| | | | | | | | | | |
|---|---|---|---|---|---|---|---|---|---|
| Apple | Rose | Road | Sea | Shark | Dolphin | Dinosaur | Crocodile | Lizard | Leopard |
| Man | Woman | Bed | Bus | Whale | Elephent | Camel | Turtle | Lobster | Caterpillar |
| Boy | Girl | Couch | Chair | Plate | Mouse | Lion | Crab | Rabbit | Squirrel |
| Baby | Tank | Tracktor | Table | Bowl | House | Tiger | Spider | Kangaroo | Seal |
| Ray | Train | Motorcicle | Bridge | Cap | Lamp | Butterfly | Orchid | Skunk | Otter |
| Can | Plane | Bicycle | Cattle | Clock | Rocket | Sunflower | Palm Tree | Pine Tree | Lawn Mower |
| Snake | Mountain | Bottle | Castle | Cloud | Trout | Tulip | Oak Tree | Willow Tree | Streetcar |
| Worm | Wolf | Bear | Forest | Mushroom | Poppy | Telephone | Maple Tree | Flatfish | Skyscraper |
| Snail | Fox | Beaver | Hamster | Shrew | Sweet Paper | Wardrobe | Aquarium Fish | Cockroach | Orange |
| Beetle | Bee | Raccoon | Possum | Porcupine | Pickup Truck | Television | Keyboard | Pear | Chimpanzee |

Figure 7: ***PoCO ordering:*** Output of PoCO for CIFAR-100, classes are separated into semantically meaningful classes (e.g., trees, humans, vehicles). We colored semantically related clusters manually

the most compressed distilled dataset, aligning with the core vision of dataset distillation. To complete the analysis, we conducted additional experiment, evaluating PoDD and RaT-BPTT on all common IPC settings as can be seen in Tab. 6. Overall, we can see that the poster becomes more beneficial as the number of IPC decreases. This makes sense, as methods must make very efficient use of the distilled images for low IPC, while this is less critical for high IPC.

# 6  Discussion and future work

Beyond the exciting result of distilling a dataset into less than one image, PoDD presents a new setting and distillation representation that opens up new and intriguing research problems.

**Class ordering algorithm.** As shown in 5.3, the ordering of the classes within a poster is strongly correlated with the performance of the distilled poster. We proposed PoCO, a greedy algorithm for choosing a class ordering. In Fig. 7 we show an example ordering for CIFAR-100, as can be seen, the classes are separated into semantically meaningful classes (e.g., trees, humans, vehicles). However, PoCO does not always yield a perfect ordering, e.g., the leopard may not fit well in the top right corner. It might fit better next to the lion and the tiger (4 left and 2 down). Indeed, other ordering strategies may be better suited for the distillation task, e.g., a photometric-based ordering that uses the color of the images or an ordering that uses the image semantics. However, computing such an ordering based on tens of thousands, or even millions of images may be costly. In contrast, our label-based approach requires a single sample per class, and can thus scale to very large dataset. Further investigation of alternative ordering methods is left for future work.

**Patch augmentations** In this work we use the extracted patches with no modifications. However, performing spatial augmentations (e.g., scale, rotation) during the distillation process may be beneficial. Another option is to create a cyclic poster where patches near the border of the poster are wrapped around.

# 7  Conclusion

In this work, we propose poster dataset distillation (PoDD), a new dataset distillation setting for tiny, less than 1 image-per-class budgets. We develop a method to perform PoDD and present a strategy for ordering the classes within the poster. We demonstrate the effectiveness of PoDD by achieving on-par SoTA performance on the well established 1 IPC benchmark with as low as 0.3 IPC and by setting a new 1 IPC SoTA performance for CIFAR-10, CIFAR-100, and CUB200.

# 8  Acknowledgments

This work was supported in part by the "Israel Science Foundation" (ISF), the "Council for Higher Education" (Vatat), the "Center for Interdisciplinary Data Science Research" (CIDR), the "Israeli Cyber Authority", and "KLA".

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

# A    Appendix

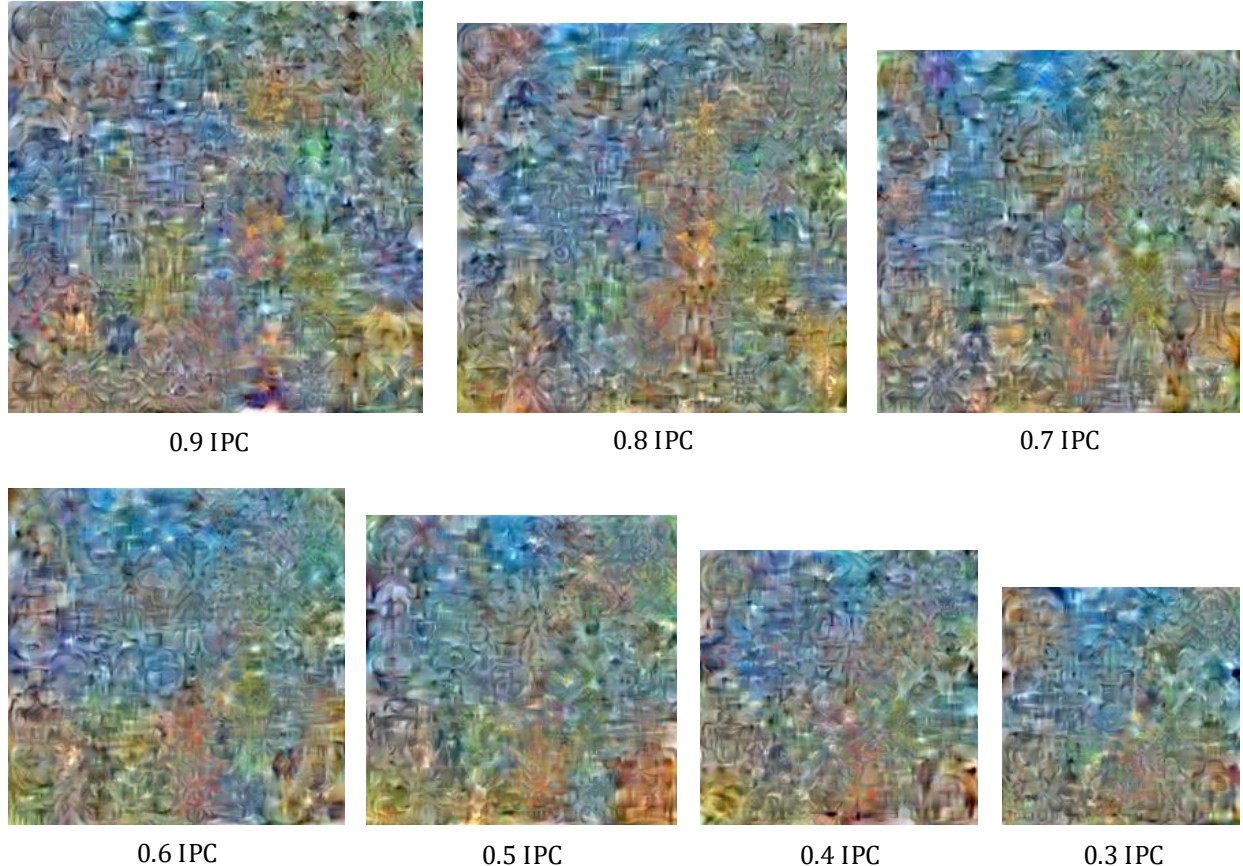

Figure 8: ***Poster visualization:*** Here we provide the poster visualization of all Less then 1 IPC settings for CIFAR100. It can be seen that the overall structure of the poster is similar, while in larger IPC settings there are more details for each general area.

**Posters size:** The poster size and label grid size used for distilling the posters in the paper.

| Dataset | CIFAR-10 | CIFAR-100 | CUB200 | T-ImageNet |
|---------|----------|-----------|--------|------------|
| $o_x$ | 5 | 10 | 20 | 20 |
| $o_y$ | 2 | 10 | 10 | 10 |
| $d_x$ | 160 | 320 | 640 | 1280 |
| $d_y$ | 64 | 320 | 320 | 640 |

Table 3: **size hyperparameters**

**Uneven class distributions:** We tested how does PoDD handle class imbalance by removing some of the samples in CIFAR-10 so the class distribution will be uneven. The resulting amount of images from each class in our experiment are: [5000, 4750, 4500, 4250, 4000, 3750, 3500, 3250, 3000, 2750]. We did not modify the test set. The resulted test accuracy compared to RaT-BPTT, as seen in Tab. 4. We can clearly see that PoDD handles the uneven dataset better then RaT-BPTT, and that the learned labels also have a positive effect.

| Method | RaT-BPTT | PoDD - Learned Labels | PoDD - Fixed Labels |
|---|---|---|---|
| Accuracy (%) | 46.8 | 56.8 | 55.7 |

Table 4: **Uneven class distributions ablation:** Test results of PoDD compared to RaT-BPTT on an uneven subset of CIFAR-10

**Downstream tasks train time** While it may appear that the larger number of patches increases training cost, we demonstrate that it does not do so in practice. Indeed, our model may converge in significantly fewer epochs than baseline models. We sometimes find that using fewer epochs increases accuracy due to less overfitting. We report our accuracy as a function of training epoch below. Note that the baseline RaT-BPTT used a set amount of 2000 epochs, which for CIFAR10 IPC1 results in 20,000 steps, to be stepwise equal we need to lower the number of epochs accordingly. Below there are tables that show the effect of lowering the number of epochs for each dataset. Tab. 5 shows the results for all IPCs settings reported in the main paper.

| Method | IPC | CIFAR-10 | | CIFAR-100 | | CUB200 | | TinyImageNet | |
|---|---|---|---|---|---|---|---|---|---|
| | | Adjusted | Full | Adjusted | Full | Adjusted | Full | Adjusted | Full |
| | | | | | **Runtime** | | | | |
| RaT-BPTT | | $\sim$ 24 h | | $\sim$ 48 h | | $\sim$ 48 h | | $\sim$ 48 h | |
| PoDD | | $\sim$ 24 h | $\sim$ 36 h | $\sim$ 48 h | $\sim$ 72 h | $\sim$ 48 h | $\sim$ 72 h | $\sim$ 48 h | $\sim$ 96 h |
| | | | | | **Accuracy** | | | | |
| RaT-BPTT | 1.0 | 53.2 | | 35.3 | | 13.8 | | 20.1 | |
| PoDD | 1.0 | 59.43 | 59.11 | 38.15 | 38.26 | 16.45 | 16.24 | 19.31 | 19.96 |
| PoDD | 0.9 | 58.27 | 58.32 | 37.18 | 37.16 | 15.58 | 14.99 | 18.63 | 19.15 |
| PoDD | 0.8 | 55.94 | 55.94 | 37.31 | 37.11 | 15.81 | 15.22 | 18.48 | 18.95 |
| PoDD | 0.7 | 54.39 | 54.48 | 37.04 | 36.82 | 15.61 | 14.91 | 18.38 | 18.51 |
| PoDD | 0.6 | 49.78 | 49.84 | 36.29 | 36.15 | 15.15 | 14.88 | 18.40 | 18.75 |
| PoDD | 0.5 | 49.49 | 49.45 | 36.04 | 36.06 | 15.18 | 14.71 | 18.03 | 18.49 |
| PoDD | 0.4 | 46.32 | 47.15 | 35.59 | 35.53 | 15.09 | 14.92 | 17.91 | 18.34 |
| PoDD | 0.3 | 42.57 | 42.13 | 34.59 | 34.42 | 14.75 | 14.44 | 17.77 | 18.13 |

Table 5: **Downstream tasks train time accuracy comparison:** Results and training times of PoDD on all 4 reported datasets, with training steps adjusted to match RaT-BPTT. As can be seen, adjusting the number of training steps does not decrease the performance, allowing PoDD to train in the same time as RaT-BPTT while increased the increased performance.

| IPC / Method | RaT-BPTT | PoDD |
|---|---|---|
| 1 | 53.2 | 59.1 |
| 10 | 69.4 | 71.0 |
| 50 | 75.3 | 75.3 |

Table 6: **Larger IPC ablation:** Test accuracy results of PoDD on the CIFAR-10 dataset using common 1-IPC, 10-IPC, 50-IPC settings compared to RaT-BPTT

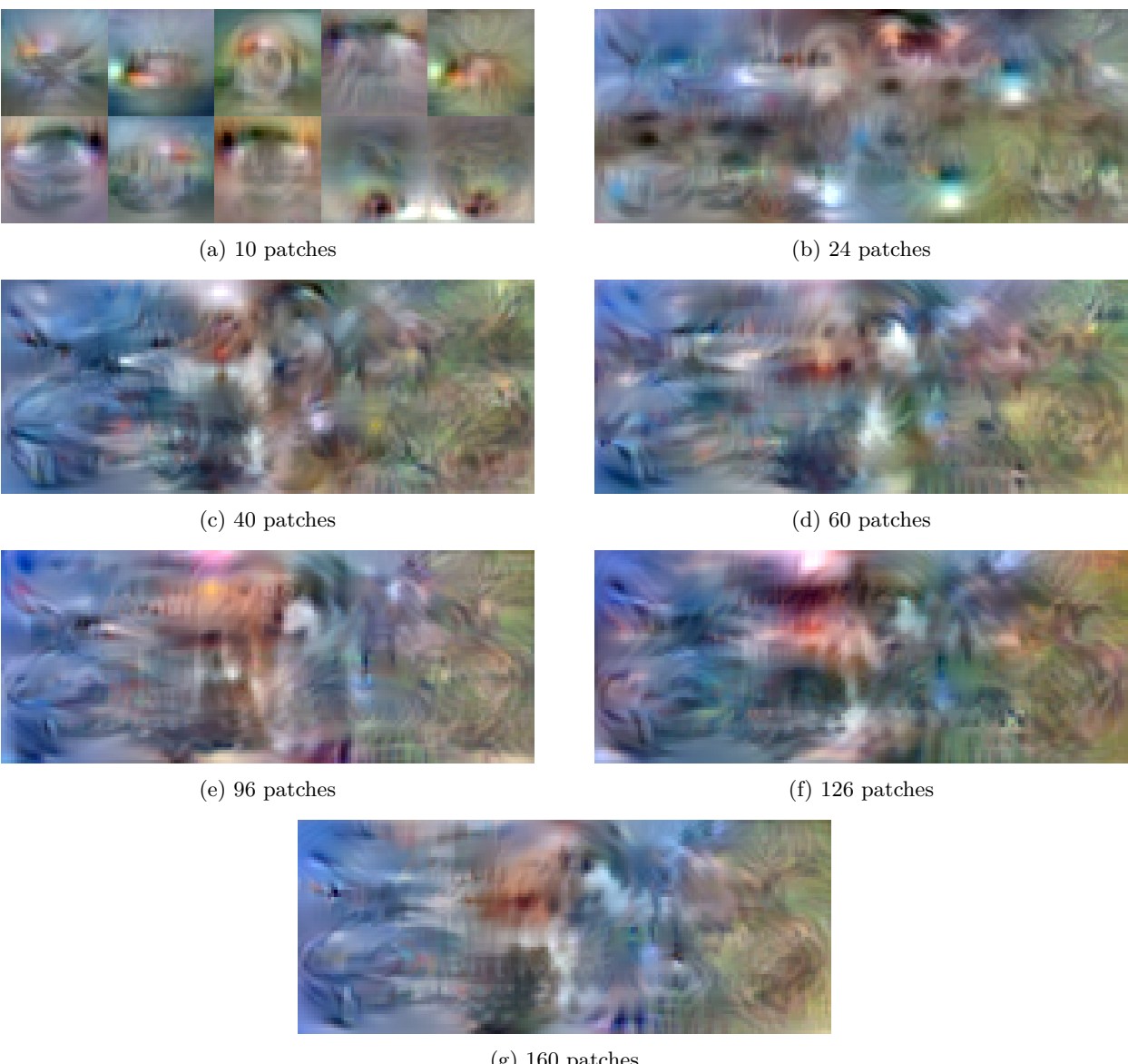

(a) 10 patches

(b) 24 patches

(c) 40 patches

(d) 60 patches

(e) 96 patches

(f) 126 patches

(g) 160 patches

Figure 9: **Patch density visualization:** Here we provide the poster visualization of CIFAR10 with different patch numbers, as seen in 6. It shows the increasing patch overlap from no overlap at all—where the patch number equals the number of classes (similar to regular distillation)—up to 16 times more patches.

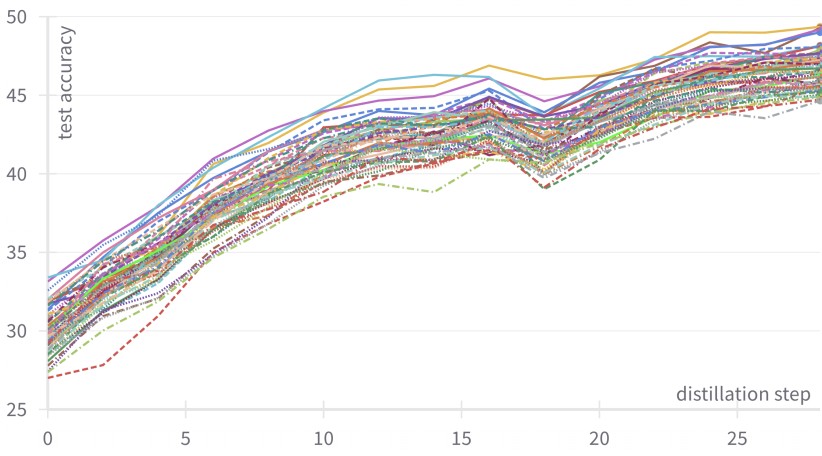

Figure 10: **Class order effect:** Using CIFAR10 as the base dataset to distill, we sampled 60 random class orders, and for each we conducted a distillation with the same amount of steps. We can see that the class ordering indeed has an effect on the test accuracy, with correlation to the ordering score.

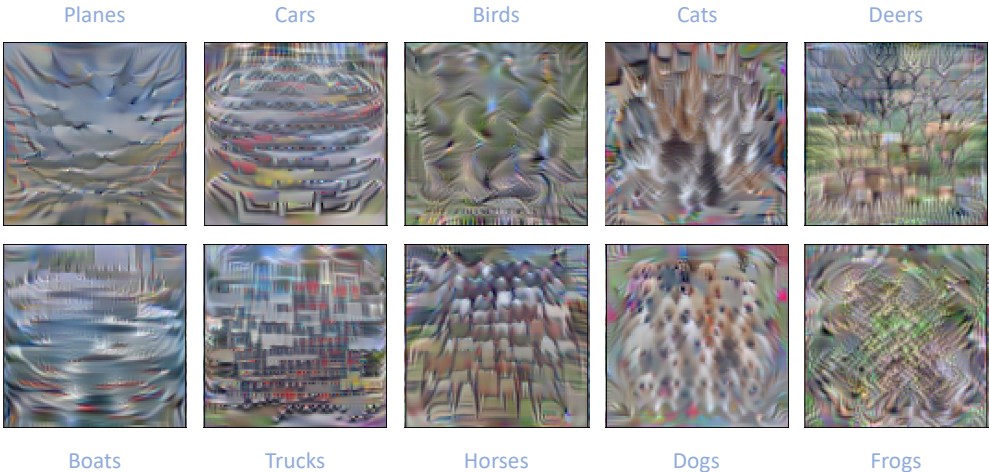

Figure 11: ***Global and Local Semantics:*** We train a CIFAR-10 variant of PoDD with 10 IPC and a separate *per class* poster. The local semantics are well preserved, showing multiple modalities per class, e.g., different colors of cars, poses of animals, and locations of the planes. Moreover, some of the classes demonstrate global scale semantics, e.g., the planes have sky on the top and grass on the bottom

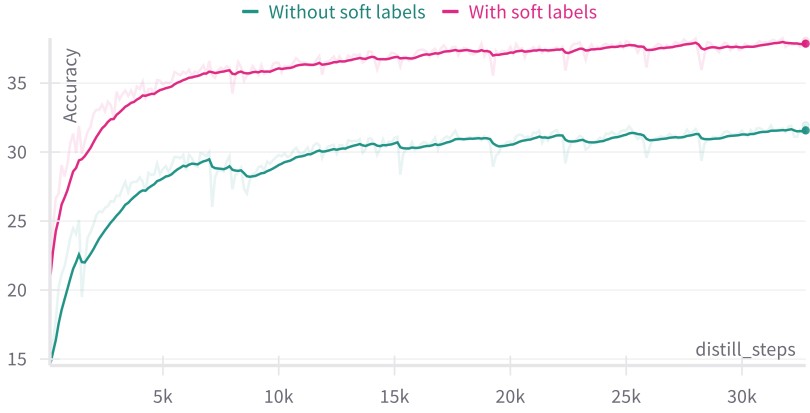

Figure 12: ***Learnable soft labels ablation:*** We ran the same experiment as the one in the main paper with CIFAR100 and IPC 1, but this time without the optimizing the soft labels alongside the image pixels. this resulted in a large negative impact on the test accuracy results throughout the distillation process

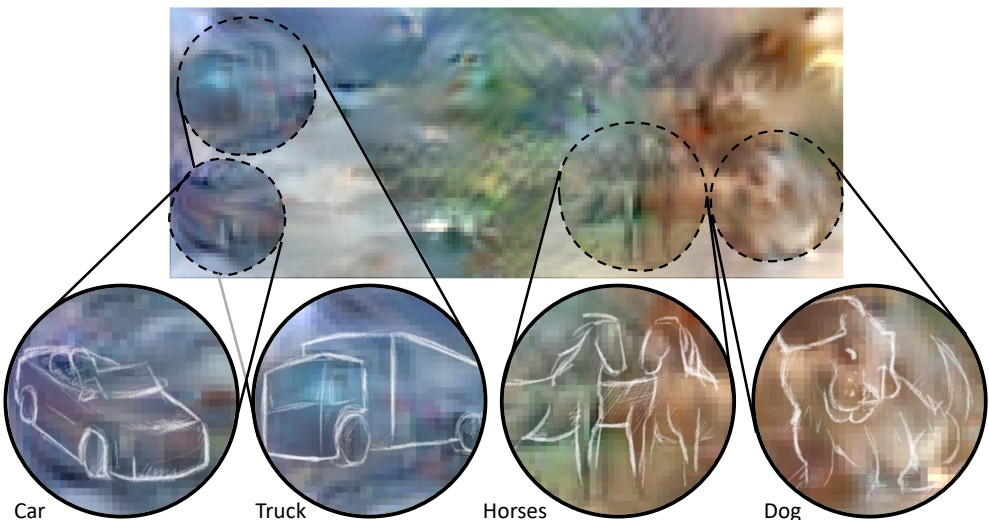

Figure 13: ***Distilled poster semantics:*** We illustrate some of the semantics captured by a CIFAR-10, 1 IPC poster by sketching over the distilled poster. The poster is of dimension $5 \times 2$, with the top row containing: Truck, Plane, Bird, Deer, Cat, and the bottom row containing Car, Boat, Frog, Horse, Dog. We can see that the pixels shared between classes exhibit a smooth transition between colors

