# OpenReview forum: "Distilling Datasets Into Less Than One Image"
_TMLR — Accepted by TMLR_

### Review · Reviewer_xPR6 · 2025-01-05

**Summary Of Contributions:**

The paper proposes a new framework called Poster Dataset Distillation (PoDD). PoDD distills a dataset into a poster. In this way, we can have less than 1 image per class for the enitre dataset.  The approach is experimentally evalauted on CIFAR-10, CIFAR-100, and CUB200.

**Audience:**

Yes

**Claims And Evidence:**

Yes

**Requested Changes:**

Please address the weaknesses above.

**Strengths And Weaknesses:**

Strengths:

1. The paper is well written and well structured.

2. The idea of distilling the entire dataset into a poster is interesting.

Weaknesses:

1. In Table 2, the SOTA works (for example DSA) seem to be using AlexNet as the network architecture. Is this consistent across both the proposal and the all the SOTA? Please add the details of the model architectures used.

2. It is important to make clear the contributions of the paper. What is the benefit of using the proposed method versus some other SOTA work? It is essential for the reader to understand the benefits of using the poster. Do we have storage savings for the training set? Does it translate to improvement in the training time? To the best of my understanding, PoDD does not offer improvements in terms of the training time compared to SOTA as eventually the same number of images are used to train the model (we are using the various patches of the poster). By using the poster, do we save in terms of storage memory? Could you please clarify this? In Table 1 maybe add a column for each dataset that shows the benefits (maybe in storage?).

3. The code is not included in the submission. Open-sourcing the code improves transparency.

4. In Table 1, for using K = 0.9 there is a drop in accuracy from 1\% till 6\% depending on the dataset compared to using 1 IPC and for smaller K the drop is higher. Could you please elaborate more on that and share any insights for this? In connection to comment 2, if the paper highlights the savings then this creates a trade-off between savings and accuracy.

5. The ablation study for the order of classes needs some more results. Please mention the accuracy of using PoCO and when the classes are ordered randomly. A table that includes this comparison would help the reader to undestand better PoCO.

6. Please add the ablation study of using vs not using learned labels for the PoDD.

7. From figure 3, we see that soft labels are used. Can you please report the experimental results when using soft labels and when not?

8. What would be the results when using the PoDD with sharing representations and without sharing representations. in this way the reader can understand better the benefit of sharing pixels between different images.

Minor comments:

1. Please add the reference to Table 1 in the manuscript.

2. In Table 1 and 2, the numbers reported after the plus and minus refer to the standard deviation over multiple runs/seeds? How many different seeds did you you use? Could you please add these clarifications?

3. Please add the references in Table 2 for the SOTA works.

4. Please mention what is the baseline that you refer to in section 5.2

5. Please add the axes titles in Figure 6

---

> ### Author Response · Authors · 2025-01-20
>
> We thank the reviewer for mentioning that “The idea of distilling the entire dataset into a poster is interesting.” Below, we provide detailed responses to the reviewer’s concerns.
>
>
> ___
> > It is important to make clear the contributions of the paper. What is the benefit of using the proposed method versus some other SOTA work? It is essential for the reader to understand the benefits of using the poster. Do we have storage savings for the training set? Does it translate to improvement in the training time? To the best of my understanding, PoDD does not offer improvements in terms of the training time compared to SOTA as eventually the same number of images are used to train the model (we are using the various patches of the poster). By using the poster, do we save in terms of storage memory? Could you please clarify this? In Table 1 maybe add a column for each dataset that shows the benefits (maybe in storage?).
>
> The main contribution of the paper is proposing the poster representation and devising different ways ( PoDDL and PoCO) to adapt current distillation methods accordingly.  The practical significance of our poster representation is twofold:
> It significantly improves the accuracy at 1 image-per-class (IPC) over the state-of-the-art results on multiple datasets.
> It provides a way to perform distillation at  less than 1-IPC. This is interesting as it achieves a higher compression ratio, which storage-starved applications may require. Specifically, we achieved better than state-of-the-art with as little as 0.3 IPC.
>
>
> ___
> > The code is not included in the submission. Open-sourcing the code improves transparency.
>
> We  added the code to the supplementary material.
>
> ___
> > In Table 1, for using K = 0.9 there is a drop in accuracy from 1% till 6% depending on the dataset compared to using 1 IPC and for smaller K the drop is higher. Could you please elaborate more on that and share any insights for this? In connection to comment 2, if the paper highlights the savings then this creates a trade-off between savings and accuracy.
>
> Reducing the number of IPCs effectively reduces the pixel budget of the distilled dataset. This results in a higher compression ratio and lower accuracy. However, note that even with fewer pixels, our method still outperforms or matches current SoTA results.
>
> > The ablation study for the order of classes needs some more results. Please mention the accuracy of using PoCO and when the classes are ordered randomly. A table that includes this comparison would help the reader to understand better PoCO.
>
> As we discuss in Sec. 5.3, we find a correlation of 0.76 between the score of the ordering and the accuracy of the distilled poster. Following the reviewers suggestions, we’ve added a figure (Fig. 10 in the appendix). which shows the effect of the class ordering on the results based on many randomly sampled orderings
>
> ___
>
> > Please add the ablation study of using vs not using learned labels for the PoDD.
>
> Following the reviewers suggestion, we’ve added an ablation to this in the appendix (Fig. 12), as can be seen there, not using learned labels has a large negative impact on the results.
>
> ___
>
> > In Table 2, the SOTA works (for example DSA) seem to be using AlexNet as the network architecture. Is this consistent across both the proposal and the all the SOTA? Please add the details of the model architectures used.
>
>
> We use the architectures and numbers reported by our baselines, in particular, a model with 3 convolution layers for 32 × 32 images and 4 convolution layers for 64 × 64 images. The exception is DSA which as the reviewer noted and as seen in their paper, uses AlexNet for CIFAR-10 and CIFAR-100.
> ___
>
>
> > From figure 3, we see that soft labels are used. Can you please report the experimental results when using soft labels and when not?
>
> As we explain in our implementation details, “We use the learned label variant of PoDDL for all of our experiments except for CIFAR-10 with $K\in[0.7, 0,8, 0.9, 1.0]$ IPC in which we use the fixed labels variant (for which learned labels did not provide additional benefit)”.
>
> ___
>
> > What would be the results when using the PoDD with sharing representations and without sharing representations. In this way the reader can understand better the benefit of sharing pixels between different images.
>
> We think there may be a small misunderstanding here, PoDD without sharing representations is exactly the same as the RaT-BPTT method, so the results would be the same as they report. Our contribution is the idea of sharing the representations using the poster.

---

> ### Comment · Reviewer_xPR6 · 2025-01-28
>
> I would like to thank the authors for the rebuttal and for uploading the code in the supplementary material. Most of my concerns have been addressed. I still have some doubts regarding:
>
> 1. "In Table 2, the SOTA works (for example DSA) seem to be using AlexNet as the network architecture. Is this consistent across both the proposal and the all the SOTA? Please add the details of the model architectures used.
> We use the architectures and numbers reported by our baselines, in particular, a model with 3 convolution layers for 32 × 32 images and 4 convolution layers for 64 × 64 images. The exception is DSA which as the reviewer noted and as seen in their paper, uses AlexNet for CIFAR-10 and CIFAR-100."  Please clarify this in the manuscript and mention the reason that this work is using a different architecture and how the proposal performs using this.
>
> 2. "What would be the results when using the PoDD with sharing representations and without sharing representations. In this way the reader can understand better the benefit of sharing pixels between different images.
> We think there may be a small misunderstanding here, PoDD without sharing representations is exactly the same as the RaT-BPTT method, so the results would be the same as they report. Our contribution is the idea of sharing the representations using the poster." Please mention this expicitly in the manuscript, as it is confusing for the reader.
>
> 3. "The main contribution of the paper is proposing the poster representation and devising different ways ( PoDDL and PoCO) to adapt current distillation methods accordingly. The practical significance of our poster representation is twofold: It significantly improves the accuracy at 1 image-per-class (IPC) over the state-of-the-art results on multiple datasets. It provides a way to perform distillation at less than 1-IPC. This is interesting as it achieves a higher compression ratio, which storage-starved applications may require. Specifically, we achieved better than state-of-the-art with as little as 0.3 IPC." It is important to clarify this in the manuscript. For the case of using 1-IPC, what are the benefits of using the poster and not some other method (that creates 1-IPC and create the poster out of this). I feel that explaining this point will make the contributions of the paper clear to the readers.

---

> > ### Author Response · Authors · 2025-01-28
> >
> > We thank the reviewer for their helpful feedback. We've uploaded a revised manuscript with the requested clarifications to the main paper, addressing all remaining points:
> > * We added architecture details to the Table 2 caption.
> > * In Section 4.1, we explicitly mentioned the no-overlap case.
> > * We made the main contributions clearer, as the reviewer suggested.
> >
> > All changes were highlighted in the revised PDF.
> >
> > We would be happy to keep the discussion open to address any further comments.

---

> > > ### Comment · Reviewer_xPR6 · 2025-01-29
> > >
> > > Thank you for the revised manuscript and for addressing the points raised in my review.

---

### Review · Reviewer_qYVG · 2025-01-06

**Summary Of Contributions:**

The paper proposes a novel distillation framework that enables dataset distillation into less than one image per class. The framework utilizes a large poster, which is transformed into patches to serve as the distilled dataset. A labeling method using CLIP embedding is proposed to designate the classes in the poster. Experimental results demonstrate the framework's effectiveness, achieving strong results even when distilling into 0.3 images per class.

**Audience:**

Yes

**Broader Impact Concerns:**

No need to discuss.

**Claims And Evidence:**

No

**Requested Changes:**

1. Provide comparisons with prior work that utilize compressed parameterization of the distilled dataset.
2. Include runtime estimates.
3. Provide a cross-architecture analysis.

**Strengths And Weaknesses:**

### Strengths:
1. The paper proposes a novel design that enables dataset distillation into less than one image per class.
2. The performance improves significantly, achieving strong results even for distillation into 0.3 IPC.


### Weaknesses and Questions:
1. The primary concern is the lack of comparisons with other relevant methods. The paper only compares approaches that optimize separate images for each class, while it overlooks several methods that use compressed bases [1, 2] for the distilled images. For example, [1] uses a smaller image basis to linearly span the distilled dataset and could effectively achieves less than 1 IPC within their framework. At that time, using integer IPCs was the standard setting and they only conducted experiments for that. However, with the current paper proposing new settings, a proper evaluation of these prior methods is necessary. Similar to PoDD, these methods leverage shared information across classes and images to reduce the parameters required for the distilled dataset.
2. How do changes to the poster affect optimization speed? Does the proposed method take longer to optimize compared to baseline methods like RaT-BPTT? The paper should include runtime estimates for all settings.
3. Use of "Optimal": The statement "It is thus important to find the optimal ordering of classes on the poster" is somewhat misleading, as PoCO is not demonstrated to generate the optimal ordering. Similarly, "optimal" is used in Section 5.3 and should be revised for accuracy.
4. For the Fixed Labels setting, what would happen if soft labels were used in the first place, with the average probability after upsampling, instead of one-hot labels? How would this approach perform?
5. In Algorithm 2, PoCO is shown to be used consistently. For PoDD with learned labels, is PoCO still used? If so, does it provide an initialization for the label tensor that is later optimized?
6. The hyper-parameters $d_H, d_W, o_x, o_y$ are not specified in the paper.
7. Figure 6 use $o_x, o_y$ for the number of patches. However, in Section 4.2, $o_x, o_y$ are defined as the label grid dimensions with $o_x \dot o_y=|C|$.
8. How does increasing the number of patches affect the visualization of the poster? Could the authors provide an analysis of how the information distribution changes on the poster as the number of patches increases? While the empirical performance presented in the paper is strong, more analysis of how information is stored when distilling into less than one image per class would enhance the paper's contribution.
9. What network is used during training? Is it a CNN? How well does a poster trained on a CNN generalize to other architectures?

---
[1] Deng, Zhiwei, and Olga Russakovsky. "Remember the past: Distilling datasets into addressable memories for neural networks." Advances in Neural Information Processing Systems 35 (2022): 34391-34404.

[2] Kim, Jang-Hyun, et al. "Dataset condensation via efficient synthetic-data parameterization." International Conference on Machine Learning. PMLR, 2022.

---

> ### Author Response · Authors · 2025-01-20
>
> We thank the reviewer for recognizing our “novel design”, and highlighting that “the performance improves significantly, achieving strong results even for distillation into 0.3 IPC.” Below, we provide detailed responses to the reviewer’s concerns.
>
> ___
>
> > runtime estimates
>
> We’ve added the runtimes to the table showing the results when adjusting the number of epochs to be the same as in RaT-BPTT. This demonstrates that PoDD converges faster and with better results than the baseline. See table 4 of the appendix. Below are the results:
> |Method|IPC|CIFAR-10 (Adj)|CIFAR-10 (Full)|CIFAR-100 (Adj)|CIFAR-100 (Full)|CUB200 (Adj) | CUB200 (Full) | TinyImageNet (Adj) | TinyImageNet (Full) |
> |-|-|-|-|-|-|-|-|-|-|
> |**Runtime**||||||||||
> |RaT-BPTT||~24 h|-|~48 h|-|~48 h|-|~48 h|-|
> | PoDD |  | ~24 h | ~36 h | ~48 h | ~72 h | ~48 h | ~72 h | ~48 h | ~96 h |
> | **Accuracy** |  |  |  |  |  |  |  |  |  |
> |RaT-BPTT|1.0|53.2|-|35.3|-|13.8|-|20.1|-|
> |PoDD|1.0|59.43|59.11|38.15|38.26|16.45|16.24|19.31|19.96|
> |PoDD|0.9|58.27|58.32|37.18|37.16|15.58|14.99|18.63|19.15|
> |PoDD|0.8|55.94|55.94|37.31|37.11|15.81|15.22|18.48|18.95|
> |PoDD|0.7|54.39|54.48|37.04|36.82|15.61|14.91|18.38|18.51|
> |PoDD|0.6|49.78|49.84|36.29|36.15|15.15|14.88|18.40|18.75|
> |PoDD|0.5|49.49|49.45|36.04|36.06|15.18|14.71|18.03|18.49|
> |PoDD|0.4|46.32|47.15|35.59|35.53|15.09|14.92|17.91|18.34|
> |PoDD|0.3|42.57|42.13|34.59|34.42|14.75|14.44|17.77|18.13|
>
> ___
>
> > Provide comparisons with prior work that utilize compressed parameterization of the distilled dataset.
>
> Following the reviewer suggestion, we provide a comparison against [1] on CIFAR-100 with the 0.3-IPC setting. PoDD outperforms [1], see table below. Nevertheless, we believe that it should be possible to combine the method presented in [1] with PoDD, we leave this for future research.
>
> Regarding [2], as far as we can tell, their lowest reported IPC setting is 10-IPC which is less relevant to our <1-IPC setting.
>
> We added citations to both [1] and [2] in the revised manuscript.
>
> |  |Ours|[1]|
> |-|-|-|
> |acc|34.7|29.3|
>
> [1] Deng, Zhiwei, and Olga Russakovsky. "Remember the past: Distilling datasets into addressable memories for neural networks." Advances in Neural Information Processing Systems 35 (2022): 34391-34404.
>
> [2] Kim, Jang-Hyun, et al. "Dataset condensation via efficient synthetic-data parameterization." International Conference on Machine Learning. PMLR, 2022.
>
> ___
> > For the Fixed Labels setting, what would happen if soft labels were used in the first place, with the average probability after upsampling, instead of one-hot labels? How would this approach perform?
>
> We agree with the reviewer that this is indeed an interesting idea. This was our initial implementation, however, we found that it performed worse than using the majority, we therefore opted to use the majority as described in the paper.
>
> ___
> > In Algorithm 2, PoCO is shown to be used consistently. For PoDD with learned labels, is PoCO still used? If so, does it provide an initialization for the label tensor that is later optimized?
>
> We use PoCO both for the learned and fixed labels. While it can provide a good initialization compared to random ordering, in cases of learned labels the improvement is less significant. We’ve added a figure with such an ablation in the appendix (Fig. 10).
> ___
>
> > How does increasing the number of patches affect the visualization of the poster? Could the authors provide an analysis of how the information distribution changes on the poster as the number of patches increases? While the empirical performance presented in the paper is strong, more analysis of how information is stored when distilling into less than one image per class would enhance the paper's contribution.
>
> We’ve added 2 figures to the appendix that explore this. In Fig. 9 we show the results of varying the number of patches for CIFAR-10 and in Fig. 8 we show the results of varying the number of patches on CIFAR-100.
>
> ___
>
>
> > What network is used during training? Is it a CNN? How well does a poster trained on a CNN generalize to other architectures? + Provide a cross-architecture analysis.
>
> In line with nearly all other distillation methods, the generalization is still within architecture rather than across architecture. Some previous works [3] used GAN-based data priors, but they do not report their same-architecture performance (we presume it is considerably lower).  Achieving high same and cross-architecture performance requires training on multiple architectures at the same time. This is theoretically possible but computationally very expensive. We consider this an exciting future direction.
>
> [3] Cazenavette, George, et al. "Generalizing dataset distillation via deep generative prior." Proceedings of the IEEE/CVF Conference on Computer Vision and Pattern Recognition. 2023.
> ___
>
> > Clarifying notation and language.
>
> We thank the reviewer for their comments, we have revised the text in the manuscript (changes highlighted in yellow).

---

### Review · Reviewer_9WSY · 2025-01-07

**Summary Of Contributions:**

The paper proposes a method that can distill datasets into less than one image for each class. The main idea is to distill a dataset into a poster that contains all classes. There are two problems: 1) How to put the classes in this poster; 2) how to assign labels to this poster. To address the first problem, the paper proposes the poster class ordering (PoCO) method, which first extracts an embedding for each class name and places the classes on a rectangle grid using a greedy method. To address the second problem, the paper proposes the poster dataset distillation labeling (PoDDL), which assign labels by using a pooling operation.

**Audience:**

Yes

**Claims And Evidence:**

Yes

**Requested Changes:**

The presentation of Section 4.3 is suggested to be revised properly to make it easier to understand.

The paper is suggested to discuss more about its practical significance.

**Strengths And Weaknesses:**

Strength:

1. The paper is interesting. It proposes a method that distill one dataset into a poster. The proposed methods are novel and technically sound.

2. The proposed method is simple and effective. It achieves impressive performance when compared to the state-of-the-art methods.

Weakness:

1. Some parts of this paper are hard to understand, such as Section 4.3. What do “fixed labels” and “learned labels” mean in this paper? The paper is suggested to provide more details for the PoDDL method. For example, what role does label array Y play?

2. What is the practical significance of this work? I think it is enough to distill a dataset into a or several images for each class. Do we need to distill a dataset into less one image?

3. There are some typos. For example, “we provide pseudocode for PoDD in 2”.

---

> ### Author Response · Authors · 2025-01-20
>
> We thank the reviewer for recognizing that our “paper is interesting”, that “the proposed methods are novel and technically sound”, and that “it achieves impressive performance”. Below, we provide detailed responses to the reviewer’s concerns.
>
> ___
>
>
> > What is the practical significance of this work? I think it is enough to distill a dataset into a or several images for each class. Do we need to distill a dataset into less one image?
>
> The practical significance of our poster representation is twofold:
>
> 1. It significantly improves the accuracy at 1 image-per-class (IPC) over the state-of-the-art results on multiple datasets.
>
> 2. It provides a way to perform distillation at  less than 1-IPC. This is interesting as it achieves a higher compression ratio, which storage-starved applications may require. Specifically, we achieved better than state-of-the-art with as little as 0.3 IPC.
>
> ___
>
> > Some parts of this paper are hard to understand, such as Section 4.3. What do “fixed labels” and “learned labels” mean in this paper? The paper is suggested to provide more details for the PoDDL method. For example, what role does label array Y play?
>
> Since our poster represents multiple classes, with shared pixels between them, we can’t simply assign a single label for the entire poster. We therefore adapt common dataset distillation labeling methods to work for PoDD. In particular, we differentiate between two strategies, fixed labels and learned labels. In the fixed approach we use a label vector for each patch in the poster based on its location, in contrast, in the learned labels approach we optimize these labels alongside the pixels of the poster. We have updated the manuscript to clarify this.

---

> > ### Comment · Reviewer_9WSY · 2025-02-06
> >
> > Thank you for your response. My concerns have been solved.

---

### Decision · Action_Editor_kGji · 2025-02-17

**Recommendation:** Accept as is

**Comment:**

No further comments beyond those highlighted in the previous sections.

**Audience:**

Yes, the proposed approach would be of interest to anyone working on dataset distillation, which includes researchers within the TMLR's audience.

**Claims And Evidence:**

Reviewers generally felt convinced by the claims made in the paper, as well as by the experimental setting. Furthermore, reviewers acknowledged the novelty and simplicity of the proposed approach. On the more critical side, reviewers believe that the practical significance of this work should be better discussed, also stressing the potentially limited broader impact of the proposed work.